# AI and Clinical Decision Making: The Limitations and Risks of Computational Reductionism in Bowel Cancer Screening

Saleem Ameen [1,*] , Ming-Chao Wong [2], Kwang-Chien Yee [1] and Paul Turner [2,*]

1   School of Medicine, College of Health and Medicine, University of Tasmania, Hobart 7000, Australia; kwang.yee@utas.edu.au
2   College of Sciences and Engineering, Information and Communication Technology, University of Tasmania, Hobart 7000, Australia; ming.wong@utas.edu.au
*   Correspondence: saleem.ameen@utas.edu.au (S.A.); paul.turner@utas.edu.au (P.T.)

**Abstract:** Advances in artificial intelligence in healthcare are frequently promoted as 'solutions' to improve the accuracy, safety, and quality of clinical decisions, treatments, and care. Despite some diagnostic success, however, AI systems rely on forms of reductive reasoning and computational determinism that embed problematic assumptions about clinical decision-making and clinical practice. Clinician autonomy, experience, and judgement are reduced to inputs and outputs framed as binary or multi-class classification problems benchmarked against a clinician's capacity to identify or predict disease states. This paper examines this reductive reasoning in AI systems for colorectal cancer (CRC) to highlight their limitations and risks: (1) in AI systems themselves due to inherent biases in (a) retrospective training datasets and (b) embedded assumptions in underlying AI architectures and algorithms; (2) in the problematic and limited evaluations being conducted on AI systems prior to system integration in clinical practice; and (3) in marginalising socio-technical factors in the context-dependent interactions between clinicians, their patients, and the broader health system. The paper argues that to optimise benefits from AI systems and to avoid negative unintended consequences for clinical decision-making and patient care, there is a need for more nuanced and balanced approaches to AI system deployment and evaluation in CRC.

**Keywords:** artificial intelligence; machine learning; patient outcomes; socio-technical design; algorithmic bias; clinical interaction

## 1. Introduction

In late 2016, Geoffrey Hinton, arguably one of the most influential researchers in the field of ML and pioneer of neural network architectures and deep learning, evocatively exclaimed that the technology was so profound that "if you work as a radiologist, you are like the cayote already over the edge of the cliff that hasn't yet looked down . . . people should stop training radiologists . . . it's just completely obvious that within five years deep learning is going to do better than radiologists"; a view that was solidified in a published opinion piece that spoke about how deep learning would fundamentally transform health care as we know it [1,2]. It has been five years since those remarks and clearly AI has not replaced radiologists. To the contrary, radiologists are in higher demand than ever-before [3]. If we trace the long history of healthcare information technologies (HIT), such as the electronic health record (EHR), computerised physician order entry system (CPOE), computer aided decision support system (CAD), e-prescription software, and now AI-enhanced HIT, the narrative has always been the same, that computational systems are introduced with promises of "computational superiority" to the benefit of patient outcomes, by enhancing the safety, quality, personalisation, and efficiency of healthcare services.

However, this hyperbole has been constantly challenged by an extensive body of research in the last three decades that has repeatedly highlighted how HIT results reported

in experimental settings are rarely emulated in the real-world of clinical practice. There is also a considerable and growing body of evidence highlighting negative unintended and unanticipated consequences (NUCs) arising at the interface between new HIT tools and socio-organisational systems, leading some to suggest that HIT seems to foster the creation of errors rather than reduce their likelihood [4,5]. These circumstances are no less prevalent in the era of AI, where overly optimistic descriptions of AI systems tend to marginalise possible risks associated with their implementation in real-world practice, particularly when system development is challenged by the nature of contextual and socio-organisational factors in the practice of diagnostic and therapeutic clinical decision-making for the delivery of safe, high quality patient care. This has not prevented technology vendors and AI advocates continuing to promote the view that healthcare is ready for disruption by AI-enhanced HIT and reiterating the standard computational narrative that AI will be the panacea for all problems plaguing the healthcare system. These include but are not limited to problems of misdiagnosis, health costs, time-scarcity, system efficiency, and treatment reproducibility.

Colorectal cancer (CRC) is currently the second leading cause of cancer and cancer-related mortality in the world [6], and therefore there has been significant interest by several IT vendors to promote clinical decision support systems (CDSS) powered by artificial intelligence (AI) and machine learning (ML) to address a subset of discrete challenges in the screening, diagnostic, and therapeutic pathways of clinical practice to improve patient outcomes. These have included preventing missed diagnoses of polyps in CC [7–9], improving reading time efficiency in CE [10], minimising interobserver variability and reproducibility in the histopathological examination of CRC [11,12], or introducing risk-stratification and prognoses prediction for CRC screening or diagnosis more broadly [13–15]. While current evidence suggests these AI tools may help address some of the problems that exist in the application of CRC screening and diagnostic technologies, it has also emerged that these AI enhanced systems themselves have their own limitations that require greater attention, if we are to avoid replacing one set of problems with another set [16]. In particular, these systems are founded on problematic representational, temporal, and cultural biases embedded in end-end data pipelines used to train AI algorithms that become further constrained by the epistemological and ontological limitations inherent to the nature of AI computation that tends to discretely frame problems of the real-world independent of context. Given the tendency for AI system developers to prescribe over-confidence in quantitative metrics produced within experimental settings, this paper explores the potential risks that arise in the absence of adequate socio-technical evaluations of AI system integration across context-dependent clinical interactions that may challenge AI efficacy across socio-cultural and socio-organisational contexts.

Of course, it is acknowledged that technology does have an important role to play in healthcare and that some forms of AI systems do support clinical decision-making and practice. However, this paper highlights how it is very important to recognise that most contemporary AI systems rely on forms of reductive reasoning and computational determinism that embed problematic assumptions about clinical decision-making, which may be problematic depending on the context of clinical practice. It is in the context of these concerns that this paper examines the opportunities and limitations of AI in CRC screening, diagnosis, and treatment. Furthermore, this paper explores whether investment in AI augmentation of CRC diagnostic modalities is misdirected, given that the majority of HIT diagnostic tools that currently support clinicians in the early detection and diagnosis of CRC, such as the Immunochemical Faecal Occult Blood Test (iFOBT) screening test, and diagnostic imaging modalities, such as conventional colonoscopy (CC), CT colonography (CTC), and capsule endoscopy (CE), have done little to change the fact that CRC remains to be the second leading cause of cancer-related death in the world, despite being one of the most preventable diseases [6].

## 2. Materials and Methods

This paper provides a socio-technical analysis of contemporary research into the use and impact of AI in colorectal cancer (CRC). This analysis is developed through a multi-disciplinary selective review, which identified over 120 papers published post-2018 through medical databases PUBMED, EMBASE, BIOMED, and Cochrane; computer science databases ACM Digital and IEEE Explore; and grey literature through Google Scholar. This approach is adopted in a similar capacity to recently published MDPI research [17,18], in order to offer a balanced critique on the opportunities, limitations, and risks of AI system development and integration in clinical decision-making along CRC screening, diagnostic, and treatment pathways.

The rest of this paper is divided into six main sections. (1) The opportunities presented by AI in health and the problematic assumptions that underpin approaches to AI development, implementation, and evaluation in healthcare are discussed. (2) The limitations and risks of these assumptions are then considered in the context of CRC, through an examination of the multi-faceted dimensions of underlying algorithms in (a) AI and data and (b) AI and models. (3) The social, legal, and ethical implications of these assumptions for AI-mediated clinical decision making are then identified and discussed. (4) Beyond these direct impacts, the paper also considers the broader impact that marginalising socio-technical factors in CRC may have on misdirecting clinical focus in ways that do little to improve patient population outcomes. (5) The paper also briefly looks to the future development of AI systems and the challenges facing regulators and practitioners in responding to the prospect of an era of 'unexplainable' AI. (6) The paper concludes by outlining steps towards building a more nuanced and balanced approach to the deployment of new AI tools in CRC to mitigate the risks to clinicians and patients in CRC diagnosis and treatment pathways. The paper points to the need for greater clarity around policies and procedures for AI clinical system validation encompassing four themes: (a) transparency and auditing of datasets, (b) transparency and reproducibility of algorithmic methodologies and implementations, (c) reproducibility of quantitative metrics through rigorous testing standards across diverse population distributions and under-represented edge cases that are known to challenge AI reliability in clinical practice, and (d) a robust socio-technical framework for AI system evaluation using a systems based approach that is sensitive to (i) the impact of AI system integration on patient outcomes, (ii) the clinical utility of AI system development for health, (iii) the nature of AI integration in the context to existing human computer interaction (HCI) and workflow considerations, and (iv) the nature of clinical interaction with and without AI.

To identify papers, the authors searched key terms relevant to the six sections that included: artificial intelligence, machine learning, deep learning, medicine, health, colorectal cancer, bowel cancer, screening, and participation. Duplicate papers were removed, and subsequent papers were screened according to relevance, application, citation score, and year of publication. Recent research from the last five years was prioritised for the selective review (*n* = 126 papers). However, historical seminal works were still considered where appropriate. Review papers and opinion pieces were excluded from formal analysis and discussion. However, they were referenced if it provided more context for the reader. Only papers with published results were included in the final analysis. In total, 88 peer-reviewed papers form the main analysis and discussion of this paper.

## 3. AI in Health: Opportunities and Concerns for Clinical Decision Making

AI in health is a rapidly developing field that has recently demonstrated a remarkable capacity to learn interrelationships within data in a way that offers utility for clinical decision making [19–22]. Most of the recent AI optimism for healthcare has been driven by the phenomenal development and success of deep learning (DL) [21–23]. In this section, we briefly review the different supervised, unsupervised, and reinforcement DL approaches [24,25] that have been used across the screening, diagnostic, therapeutic, and prognostic pathways of clinical decision making. In navigating the application of these

methods in medical practice, we introduce some of the problematic assumptions that are embedded in AI system development. Some of the problems are general in nature and implicit to the process of AI system development (e.g., the biases embedded in data used to train AI systems), while others are specific to certain AI algorithmic methodologies that become more visible in the healthcare space of CRC. The purpose of this section is not to undertake an extensive review of AI methodologies, but to highlight that there are inherent methodological considerations that underpin their development that impact on clinical decision making. This indicates that a more nuanced approach to examining AI in CRC is necessary if we are to optimise AI benefits and limit its harms in clinical practice.

The medical literature is ripe with examples that demonstrate the highly efficacious modelling capacity of supervised ML across myriad medical modalities. These have included predicting: (a) benign/malignant cancerous states from pixel data found in photographs of skin lesions [26]; (b) classifications of disease states in chest radiographs (e.g., normal vs. abnormal radiographs, presence of pneumonia, presence of malignant pulmonary nodules) [27–29]; (c) risk stratification and prognosis from whole slide images (WSI) of histopathological tissue specimens that are used to inform therapeutic pathways in areas such as gastroenterology [12,13,30]; (d) arrythmia, atrial fibrillation, or coronary heart disease from wave data in electrocardiograms [31–33]; (e) the likelihood of sepsis based on clinical observation notes and test results found in EHRs [34]; (f) the presence or future onset of neurological diseases, such as brain tumours or Alzheimer's disease, from CT, MRI, or positron emission tomography [35–37]; (g) cardiovascular risk from fundus photography [38]; and (h) colonic polyps in colonoscopy [39], among innumerable other examples [40,41]. Despite the success of supervised ML across myriad medical contexts, researchers have started to identify that there are complex nuances that underpin (a) AI and data, and (b) AI and models, that pose significant challenges to supervised learning systems when modelling the heterogeneity of the real-world, due to its dependency on large volumes of labelled data and narrow task definitions framed by the human observer.

One of these issues includes representational harms introduced through data collection and/or labelling practices [42,43]. For example, in 2019, a landmark study was published in *Science*, where Obermeyer and colleagues [44] revealed how systemic racial biases emerged after auditing a proprietary ML algorithm used routinely on 200 million people in the United States of America each year that was tasked with assigning risk scores on patients that would be eligible for subsided "high-risk care management". While the algorithm was well intentioned, the elusive nature of systemic racial biases meant that the AI was unable to recognise that generational inequalities in healthcare access between two sociodemographic groups resulted in a situation where less money was spent caring for less-healthy Black patients compared to more-healthy White patients. Therefore, at a given risk score, Black patients were considerably more ill than White patients. Interestingly, re-labelling the data with a proxy variable that combined current health status with expenditure was shown to reduce racial bias by 84% and increase the percentage of Black patients receiving additional care from 17.7% to 46.5%.

Beyond systemic biases, another issue that arises is that AI generalisability is dependent on diverse and equitable representation in data distributions, which may not always be achieved by big data. As some genomic studies have shown, marginalised groups may be underrepresented in the data, and their use in practice may lead to confounders and incorrect correlations, as was seen in the prediction of hypertrophic cardiomyopathy in Black versus White patients [45]. Yet, there are public genotype repositories such as the 23AndMe dataset, which is based on 87% European/White representation and only 2% Asian and Black representation respectively [46], and the UK Biobank has a "healthy volunteer" selection bias [47]. Both of these tend to form the basis of training published ML algorithms [48,49]. This could limit the success of the algorithms when used on broader patient cohorts. While there are approaches to mitigate this issue by balancing the data through under/over-sampling techniques and data augmentation [50,51], an accuracy paradox may emerge, since the quality that enabled the AI to perform efficaciously on

the dominant class initially (prevalence of large volumes of that class) can be conflated by the prevalence of the newly augmented class. Furthermore, the class that was augmented may have contentious validity, as the nature of synthetic data generation may mean that generated samples do not actually represent the disease state in question [52]. For instance, a chest radiograph that experiences a horizontal flip augmentation can inadvertently result in the depiction of a different medical condition called situs inversus.

More broadly, other issues that have been reported on in the literature include (a) interobserver variability among labellers, which can limit the veracity of the data used to train AI models [53], (b) challenges in delineating the suitable level of abstraction when labelling disease presentations in a way that morphologists agree with [54], and (c) intrinsic biases introduced by the contentious validity of narrow task definitions pre-imposed by the human observer in framing a reductive computational relationship that is maximised between an input $X$ and an output $Y$ independent of broader interrelationships between clinical history, examination, and interaction [55].

Contemporary unsupervised learning and reinforcement learning approaches have attempted to deal with some of these problems in healthcare by approaching computation in a way that is more emblematic of human approaches to knowledge discovery and decision making [56–58]. Unsupervised learning has shown that a data-driven approach independent of human interference can successfully disentangle meaning out of complex data structures, such as learning implicit brain MRI manifolds to enable for better quantitative analysis and observations about the presence and/or development of disease [59], or deriving general-purpose patient representations from electronic health records to predict the onset of future disease states across diverse clinical domains and temporal windows [60]. Similarly, "goal-directed" reinforcement learning has shown much promise in the ICU setting, where optimising decision making for a longitudinal goal that requires extensive personalisation (e.g., patient survival) is highly desired. For example, one study explored how a RL algorithm could use available patient information to define a personalised regime for sedation and weaning off ventilator support in an ICU, by predicting the optimal time to extubation in a way that minimised complications arising from either (a) prolonged dependence on mechanical ventilation on one extreme or (b) premature extubation that requires reintubation on the other extreme [61]. In another study, it was demonstrated that continuous state-space models could learn clinically interpretable treatment policies that could aid ICU physicians in treating septic patients, in a way that improved the likelihood of patient survival [62].

While such studies present an important stride forward in computational clinical decision support that more closely resembles human behaviour, the significance of the results should not be overstated. As Liu et al. [58] highlight in their comprehensive review of RL algorithms developed for critical care settings, while these approaches do mitigate against some of the biases introduced by the human observer in supervised learning approaches, the methods do little to address the fact that data collection itself, may remain biased since (1) the state space used in RL systems are constructed from data constrained by the selection of patient demographics, laboratory tests, and vital signs present in the data; and (2) the task being optimised for is still defined a priori, which means, as in supervised learning, the efficacy of the system is still heavily influenced by the human observer who decides what goal should be optimised for. Most importantly, both unsupervised and reinforcement learning approaches introduce new, more complicated problems around model evaluation in the absence of a labelled benchmark, which has led to researchers such as John Kleinberg to famously declare that unsupervised clustering is so problematic that it's possible to define an "impossibility theorem" for it [63,64].

Given the importance of data and their role in shaping the efficacy of AI predictions, this paper advocates for greater collaboration between the ML, health informatics, and clinical communities to develop a standardised systems-based approach to AI evaluation prior to clinical integration and posits that datasets and algorithms should be thoroughly audited prior to integration into clinical practice. For this to work, transparency from

commercial vendor-locked systems and adequate prospective studies must become the norm. Unfortunately, a recent review of 130 FDA-approved AI medical devices found that 126 systems only ever used retrospective studies to report their results, and none of the 54 high-risk devices were evaluated by prospective studies [65]. Unsurprisingly, when one of the commercial algorithms that was being used for the detection of a pneumo-thorax was prospectively evaluated across ethnically diverse population groups, there was a statistically significant difference in performance on the algorithm's ability to accurately predict the pathology in Black versus White patients [65]. Problematically, one meta-analysis also discovered that in 516 publications highlighting the accuracy of medical AI systems, only 6% were externally validated [66]. This may suggest that there is over-optimism on the promise of medical AI and haphazard consequences may arise if sufficient external evaluations of the impact on patient outcomes in varied socio-organisational settings prior to clinical integration do not occur [67].

In the next section, we will deepen our examination of the issues of algorithmic bias in clinical decision making in CRC from a perspective of (a) AI and data, and (b) AI and models. The section integrates a perspective on how socio-technical interactions at the interface of clinical practice in CRC may marginalise the opportunities that AI may provide clinicians, patients, and the healthcare system.

## 4. AI in CRC: Limitations and Risks of Algorithmic Bias in Clinical Decision Making

Although AI introduces opportunities for clinical decision making, Figure 1 highlights how there are a range of issues that amalgamate to limit model efficacy in the real-world, the most significant of which are related to the interrelationship between: (1) AI and data, through the underlying biases present in data distributions used during model training; and (2) AI and models, through the reductive computational assumptions that emerge out of the translation of medical problems into narrow computational task definitions that are independent of context and that are constrained by limitations in underlying model assumptions. This section examines these issues in the context of CRC screening and diagnosis.

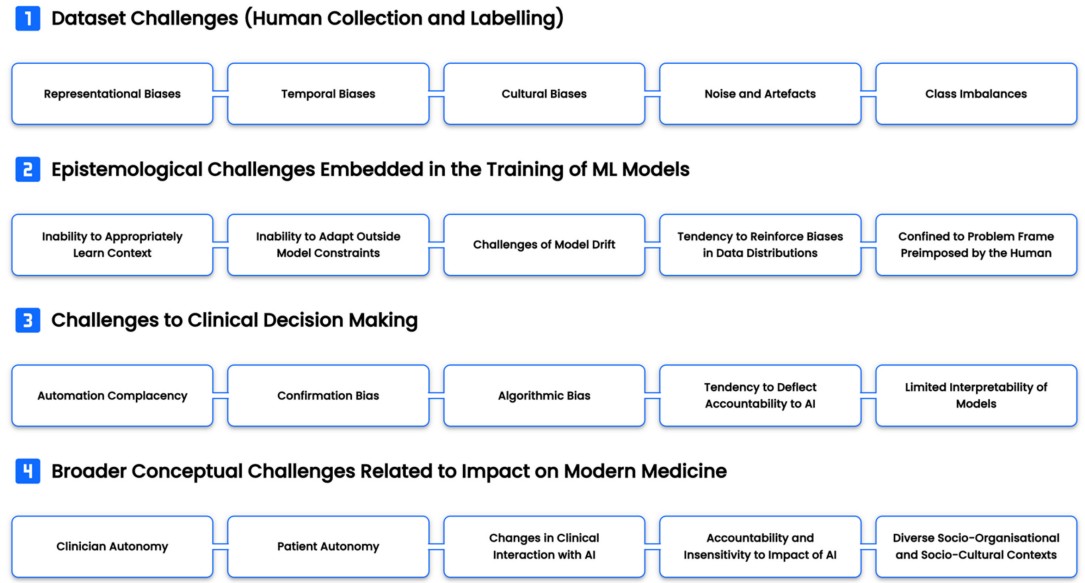

**Figure 1.** This infographic highlights the range of potential challenges associated with AI in clinical decision making and reinforces the need for a robust framework to AI evaluation prior to clinical integration, to maximise confidence in the safety and equity of system use in practice.

### 4.1. CRC, AI and Data

It is well established that data has the most significant impact on developing efficacious AI models that are robust, performant, fair, safe, and scalable across contexts [68–70]. Access to large-scaled labelled data during model training is so significant that Sun et al. [71] demonstrated that AI model performance increases logarithmically relative to the amount of training data available. However, 'big data' and 'balanced data' are not synonymous. In this section, we explore (1) representational biases and stereotypes that emerge in data relating to race, gender, ethnicity, religion, disability, sexual orientation, and socioeconomic status that alter what AI systems learn, and discuss how they may marginalise patients from backgrounds that tend to be at the highest risk of CRC and have the poorest outcomes [72–78]. Furthermore, we discuss how (2) class imbalances associated with the representation of heterogenous and/or underrepresented disease states in datasets may lead to problematic outcomes for patients with rare diseases, particularly when clinician automation complacency and bias in the presence of AI influence clinical decision making [50,79,80].

#### 4.1.1. Representational Biases in Data and CRC Risk Stratification Algorithms

In CRC screening, it is well known that the patients who participate the least in screening, who have the highest risk of CRC, and who present with the poorest outcomes, tend to be concentrated in groups that have experienced the most social disadvantage, such as people from (a) Indigenous populations, (b) low socio-economic status, (c) diverse cultural backgrounds, or (d) with disability [68–70]. For example, African Americans have the highest incidence of CRC of all ethnic groups in the United States (US), have a mortality rate that is approximately 20% higher than White Americans, and typically have a younger onset of the disease [68]. Yet, such groups tend to either be (a) underrepresented in historical machine learning datasets, or (b) when they are represented, experience algorithmic bias due to systemic inequities embedded in the nature of the data. This poses potential risks to patient care when an algorithmic prediction of patient risk and/or prognosis is clinically implemented to guide clinical decision making around who will benefit from access to treatment interventions.

These representational issues combined with a lack of a standards-based approach to medical AI evaluations poses some interesting challenges in CRC screening, particularly when there is a significant interest in developing ML methods for the screening of early-stage CRC, given that the current two-tiered "Gold Standard" FOBT + colonoscopy approach is challenged by a problem of low patient participation in screening [81]. Wan et al. [82] suggested leveraging ML with whole-genome sequencing of plasma cell-free DNA and demonstrated that is possible to predict the early onset of the disease with a mean area under the curve (AUC) of 0.92 and sensitivity and specificity of 85%. One perspective is that this whole genome approach allows for an unbiased discovery of signals that are not disease specific and can be extended to the monitoring of non-disease states through the detection of biomarker correlations. However, as the authors acknowledge, demographic and institutional biases may impact on the generalisability of the results and the need for prospective trials is emphasised. The issue with prospective trials that are not sensitive to these representational issues, is that they can be designed in a way that unwittingly supports the hypothesis, even when the authors are well intentioned. For example, Chan and colleagues [83] proposed an ensemble ML algorithm to predict recurrence of CRC using historical genomic data from a French population and claimed a sensitivity of 91.18%, which was validated on data from Australia (91.67%) and the United States of America (80%). There are concerns that the AI is therefore optimised for patients with Caucasian ancestry, which could lead to problematic outcomes if these same algorithms are then inappropriately used on patients of different ancestry that they are not optimised for. This is of particular concern when we consider that (a) patients who are often at the highest risk and who demand the most urgent care are often the ones that are least represented in

the data, and (b) that clinical decision making, according to some research, can be heavily influenced by the presence of algorithms [15,79,80].

To understand the effects that these algorithms have on clinical decision making, we point to a fascinating study published in *Nature*, where Kostopoulou and colleagues [15] setup an experiment to observe how recommendations for referral to specialised oncology care would change in 157 general practitioners (GP) from the United Kingdom (UK) when presented with 20 vignettes of patients with symptoms that might indicate potential CRC and an unnamed algorithm predicting each of the patient's risk. The researchers observed two things: (1) after receiving the algorithm's estimate, a GP's inclination to refer for specialised care changed 26% of the time, with the greatest impact seen when the GP felt that they underestimated the risk compared to the algorithm; and (2) with continued use, there was a positive GP disposition towards the algorithm, as GPs became better calibrated to the probabilistic way that the algorithm would associate symptoms with risk that they started to inadvertently emulate the same algorithmic approach to deduction and conclusion.

While this study was celebrated as a success with high clinical utility, as GPs seemed to improve their cancer referral decisions to the benefit of the patient, particularly when the AI risk predictor was higher than the clinician; what the study does not thoroughly investigate is the inverse scenario of what happens when patients who are actually high risk are provided with low-risk scores, as witnessed in the previous Obermeyer study [44]. Even though the study does seem to indicate that GPs tend to err on the side of caution and so an incorrect low risk algorithm theoretically should not change a GPs disposition to refer; the effects of these interactions have not been thoroughly investigated. Human factors engineering suggests the opposite may occur, as humans may subconsciously deflect accountability to the machine [84], particularly when the effects of confirmation bias [85,86] and automation complacency [80] set in, where clinicians are believed to lean towards the decision of an automated system and subsequently stop searching for any further confirmatory evidence. This phenomenon has been extensively discussed for two decades in cardiology around the issue of automated electrocardiogram analysis [87,88].

This is concerning, particularly when a close examination of the literature shows examples of representational biases that are both systemic and distributional in nature unwittingly emerge in the absence of a robust methodological framework to address these problems. For example, Nartowt et al. [14] developed an exclusively software-based screening tool for the early identification and prevention of CRC in large populations by training a neural network to classify individuals into low, medium, and high CRC-risk groups using only personal health data found in two public datasets: (1) the National Health Interview Survey (NHIS) dataset and (2) the Prostate, Lung, Colorectal, Ovarian Cancer Screening (PLCO) dataset. To maximise machine performance, the authors converted much of the dataset pertaining to race into a set of binary variables presenting attributes such as "Not Hispanic/Spanish origin", "Black/African American only", "American Indian only", "Other race", "Multiple race", and "Sex factor". However, this reductionism clearly has consequences. Demarcating all other ethnic groups into a single variable of "Other race" is a dangerous assumption, because it implies that there are no differences between all the other rich cultures across the world. Compounding the problem is the fact that systemic disparities that exist across groups due to sociodemographic context are not accounted for, even though it is already known that this can be a significant limitation to predicting accurate outcomes.

### 4.1.2. Class Imbalance, Heterogeneous Disease States, and Underrepresented Disease

Class imbalances and underrepresentation of rare disease states also presents an interesting challenge for AI in CRC, which is not easily remedied by simply accruing more data. The quality that makes ML so potentially powerful, the ability to learn patterns within data by maximising signals that reinforce distributions in the datasets, is also one that can lead to a situation where an AI optimises for features that are highly predictive of

over-represented disease states, at the expense of features that detect or diagnose under-represented rare diseases, even where computational techniques such as data augmentation and regularisation are implemented. For example, in a recent landmark study, Wang and colleagues [11] developed a state-of-the-art deep CNN that capitalised on transfer-learning and demonstrated superior performance to pathologists in the histopathological analysis of CRC tissue specimens, achieving a 0.988 vs. 0.970 AUC. The experimental setup appeared sound and resistant to algorithmic bias: (a) they used a large volume of data (170,099 patches sourced from 14,680 whole slide images, captured from >9631 patients), and (b) the patient cohorts attempted to be clinically representative, by collecting cases from multi-independent sources across China, the USA, and Germany. However, as the authors highlight in their analysis, several histological types were excluded from the study, because they were too rare and had less than a 0.5% incidence. While they acknowledged this limitation and stated that the algorithm would improve over time through the collection of more data, it is important to highlight, again, that balanced data are more important than more data, and the algorithm may remain skewed to the overrepresented class.

This may not be a problem in and of itself, as we acknowledge that rare diseases are difficult for clinicians to diagnose and that the net benefit of these systems may still be incredibly valuable when used as a second observer to ensure common cases are not misdiagnosed and/or missed on account of human error. However, a socio-technical analysis evaluating the system may weigh these benefits against the risk that over-confidence in the AI over time may alter the behaviour of clinical interaction, such that clinicians become less perceptive of signals that an AI is known to be poor at, due to complacency in the presence of the machine. This may mean that more careful consideration to HCI in the development of HIT systems that utilise AI is enacted, or specific clinician retraining around how to approach clinical practice in the presence of AI is mandated.

This may have interesting ramifications in emergent capsule endoscopy technologies, where AI has been heavily promoted to increase small bowel findings while reducing reading time through a mechanism that filters out normal findings and uses image processing techniques to merge similar images together [10]. While this optimisation has largely been lauded as a profound optimisation and improvement to the workflow through the reduced reading time, there are concerns around whether the models will be robust against all possible edge cases, particularly given the highly heterogenous ways that disease can manifest. We do not know the answer to these questions, but it does indicate that, at minimum, we need a rigorous framework around the external validation of AI, specific quantitative testing around edge cases that are expected to be a challenge for AI such as underrepresented classes, and more evaluations of clinician performance with and without the use of AI, to ensure that we optimise the benefits of AI and limit their harm when integrated into clinical practice.

### 4.2. CRC, AI and Models

In the previous section, the impact of data on AI performance was discussed and it was acknowledged that more sensitive approaches to developing transparent, representative, and equitable datasets could improve the efficacy of AI across diverse patient cohorts and limit potential harms. This section extends on that discussion, to highlight that there are broader contextual factors that impact clinical decision making that poses unique challenges for AI, when consideration is given to the fact that current AI system technology is functionally unable to transcend the epistemological and ontological assumptions embedded in the nature of model design. This tends to leave AI unaware of the nature of clinician, patient, or healthcare interactions and may result in erroneous conclusions by an AI system due to an inability to recognise and appreciate the nuances of (a) temporal context, and/or (b) situational operator context. Consequently, this section re-emphasises the dangers of clinician complacency in the presence of AI system implementation in clinical practice. This section concludes by examining how the inability to adequately explain AI model predictions exacerbates the impact that context has on AI reliability in practice.

4.2.1. Temporal Context

An issue that has been widely discussed in the literature is the problem of model adaptation in the presence of distributional data shift [89], where there is (1) a mismatch between the training data used by an algorithm at one point in time, and unanticipated and/or evolving patient/healthcare contexts that emerge at a later point in time; and (2) an inability of an AI algorithm to accurately adapt to such non-stationary clinical operational environments due to the way that an AI model frames its assumptions of the world [55]. This tends to manifest most when historical EHRs are used to train a ML algorithm and new data are later recorded and captured in the EHR, which was absent in the historical data due to the evolution of clinical practice. This leads to a problem known as model drift [84,90,91]. This is not an easy problem to solve, since historical data cannot suddenly be updated according to new knowledge, and new knowledge is often not voluminous enough to train a new algorithm efficaciously. Even though more contemporary methods allow for continuous learning via a process known as incremental learning [52], it is important to recognise that this process of adaptation is still bound within the initial constraints of the problem definition and an AI algorithm cannot epistemologically adapt outside of the computational model "frame" imposed by the engineer. Importantly, the ML community has not agreed on the best approach to handling class imbalances that emerge during incremental learning, since real-time learning may inadvertently undo any efforts used to balance initial data distributions and may skew the algorithm back too heavily in favour of diseases or population groups that present commonly.

To show how these issues manifest first-hand in a healthcare setting, Davis et al. [92] revealed how decreasing rates of acute kidney injury (AKI) led ML models to drift towards a state of over-prediction of AKI within one year of development and this had the negative unintended consequence of altering clinical decision making in a way that misdirected resource allocation and expenditure. A more extreme example of this problem can be seen in the emergence of the recent COVID-19 pandemic, which witnessed an unprecedented shift in the patient landscape of a typical emergency department (ED), where an exponential increase in ED visits for COVID-19 matched an exponential decline in acute visits for stroke and heart attacks [91]. This presents a potential issue for CRC, given that since the early 1990s, the incidence of colorectal cancer in patients below the age of 50 has nearly doubled, but this population is often not captured by most screening programs [93,94]. While it is true that the absolute number of these patients are currently small and are not necessarily included in screening pathways, it does demonstrate how disease patterns evolve over time, and this has consequences to how AI systems are developed and evaluated. Observations in younger CRC patients who present with more advanced stages of CRC have suggested that there are multifactorial genetic and environmental components that influence the nature of the underlying disease [93]. How does this then influence prognostic model performances, e.g., if the underlying biological mechanisms are discovered to have shifted over time? Therefore, it is important to consider that we need more robust guidelines around how and when algorithms should be re-trained or re-calibrated, to maintain their performance across shifting distributions to ensure that clinical decision making, which is inevitably influenced by algorithmic decisions, remains robust to evolving clinical knowledge and dynamic clinical settings.

The ramifications of temporal context on algorithmic predictability can be quite elusive, particularly when we also consider how AI integration into clinical decision making along therapeutic pathways is quite sensitive to the specific circumstances of the patient's own temporal context, independent of the broader population. For example, it was reported in one study by Jie and colleagues [95], that when IBM's Watson for Oncology (WFO) was used to provide an oncologic treatment recommendation for a colon cancer patient, the WFO did not recommend the usual CapeOX (oxaliplatin + capecitabine) treatment regimen, because WFO assumed it was unsafe for the patient due to a recent biochemical blood test that showed a creatinine clearance rate <30. However, when this was reviewed by the multi-disciplinary team (MDT), the oncologists immediately knew that this was a transient

reversible biochemical abnormality due to the treatment, which would organically recover after one week, concluding that it would be irresponsible to stop the CapeOx treatment scheme. On review, the creatinine clearance rate returned to normal as the MDT expected.

### 4.2.2. Situational and Operator Context

AI algorithms also tend to be unaware of situational context, where environmental factors may have a significant impact on the appropriateness of the predictions, which again reinforces the need for a standardised external validation framework that is sensitive to technical and socio-technical concerns. There have been countless examples in the broader ML literature to show how situationally unaware neural network signal optimisations have led to a model exploiting unreliable artefacts, confounders, or spurious cues in a training dataset to the detriment of its reliability and generalisability in new contexts. The most cited example of this problem in the literature is the classic case where a ML algorithm predicted pneumonia in a chest radiograph due to the type of X-ray machine equipment that was used, rather than any underlying features of pneumonia. By coincidence, the situational context was that patients who were most unwell, and most likely to have pneumonia, were the ones that required point of care imaging by the clinical staff and those radiographs were incidentally stamped with the term "portable" [29]. Given that it is known that an AI may maximise spurious cues to forge incorrect classifications, even at the level of a single pixel perturbation [96], there may be consequences that limit the currently reported success of AI-assisted polyp and/or CRC detection and diagnosis systems used in conventional colonoscopy, when they become more mainstream. In a recent study, Li et al. [97] highlighted the range of situations that contributed to false positive or false negative detections by an AI-assisted polyp detection system, which included: (a) when a polyp had approached the corners of frames when they were about to appear or disappear from the image; (b) when light reflections and shadows were present in the image due to bubbles arising from a patient's sub-optimal bowel preparation; (c) when edges of circular folds could be misconstrued as polyps; or (d) when images were out of focus and blurred.

This introduces an interesting socio-technical question: is the problem here due to the AI, or due to the interaction of the human observer with the AI? One could argue that the AI was performing as intended—it disentangled the factors of variation to demarcate polyps most of the time, as advertised—, but it had some expectations around how the human observer interacted with it. This could include expectations around the endoscopist's pace of movement to limit the likelihood of blur, the patient's minimal bowel preparation, or expectations around the endoscopist's approach to insufflation or irrigation of the bowel. This highlights that clinical decision making may need to be adjusted in the presence of AI and clear guidelines around the expectations and limitations of these systems in-use are needed so that we optimise their benefits and limit their harm.

At times, situational context, patient context, clinician context, temporal context, and representational biases may all interact to affect the relevance of AI decision making. In one WFO feasibility study, Liu's team [98] investigated whether patients with lung cancer who were receiving treatment in China, of Chinese descent, could be provided with treatment recommendations that were consistent with the multidisciplinary team. The authors concluded that the overall consistency was 65.8% and could have been increased to 93.2% had the WFO considered differences between Western and Chinese contexts, such as (1) differences in the presentation of genetic mutations, (2) differences in the sensitivity, tolerance, and metabolism of chemotherapeutic agents due to different physiques that influence treatment regimens between nations (e.g., concurrent vs. sequential chemoradiation), (3) differences in the availability of drugs between markets, and (4) differences between patient preferences, particularly in lieu of treatment prices and medical insurance.

There are different perspectives around why these issues arise and the significance of their impact. Some authors such as Strickland [99] argue that these issues arise due to poor data practices by IBM. However, another perspective could be that the limitation lies in the way that the AI was framed to model the data, and had it been provided with

access to the raw data with greater computational capacity, perhaps AI could propose a personalised treatment recommendation (i.e., rather than having a treatment regimen offered according to national guidelines that dismisses concurrent chemoradiation, AI could learn to optimise the dosing of concurrent chemoradiation for the smaller physique). The existence of these competing perspectives validates the position advocated by this paper, that a robust methodology to AI evaluation in healthcare is needed. Furthermore, this example also emphasises that there are clearly dangers when commercial vendors develop algorithms using multi-centre datasets that appear equitable and efficacious on local population distributions but proceed to reductively commercialise that same AI system without any due diligence or oversight to internationally different population distributions.

### 4.2.3. Interpretability of AI Models

Since AI reliability can be affected by numerous contextual factors that arise in clinical practice, EU regulators have enforced a position through the 2018 European Data Protection Regulation stating that "black box" AI algorithms that have a significant effect on users must be able to explain why a decision was made on-demand, as patients deserve the "right to explanation" [100,101]. As a precedent, this was an important and sensible step towards developing safer AI systems that were operationalised in a way that were more complementary to how clinicians are trained to weigh competing modes of evidence to the contextual circumstances of the patient. The computational community has responded positively to these concerns by enhancing AI system interpretability through integrating (a) gradient-weighted class activation mappings (Grad-CAM) [102–104], (b) replacing singular end-end classification pipelines with sequential segmentation + classification steps [105], (c) extracting highly active neurons to visualise feature detectors [106], (d) gradient feature auditing to estimate the indirect influence a feature has on a prediction [107], and (e) using a process of "deep dreaming" to understand the evolution of a network's layers [108]. However, the diversity of the methods also indicates that "interpretability" exists on a spectrum and knowing what level of interpretation is sufficient to limit the impact of AI bias in clinical practice is unknown.

## 5. Social, Ethical, and Legal Ramifications of AI Mediated Clinical Decision Making

In the context of the technical and socio-technical critiques discussed above, it is important to delve more deeply into the wider ramifications of the underlying concepts pertaining to computational biases and reductionist assumptions embedded within most contemporary clinical AI systems. Beyond NUCs at the clinical interface, this section highlights that there are potentially broader ethical and moral questions raised by the wide-spread deployment of these systems because of their potential to transform the basis for clinical decisions. We discuss how the socio-technical perspectives provided are not simply 'contextual' concerns but are more deeply grounded in the fundamental limitations and risks embedded within AI systems themselves.

Clinical decision making is not just about treating biological disease, it is about treating a patient with a unique set of psycho-social and cultural factors. It is for this reason that two critical pillars guide the practice of ethical medicine: beneficence (the need for common good and benefit) and nonmaleficence (first, do no harm) [109]. Therefore, clinical care means that even in the absence of evidence, a clinician may choose to "err on the side of caution" and order an investigation, weighing out the risks associated with the potential of a missed diagnosis against the risks of overdiagnosis to the unique circumstances of the patient [84]. AI algorithms are not sensitive to the impact that "absolute" probabilistic decisions around disease, independent of the patient's concerns has on the patient's well-being. While some may argue that this is too strong of a criticism of AI, since AI is not the one making the final decision and is merely "supporting" the physician to make their decision; as we have discussed throughout this paper, clinicians are susceptible to a range of cognitive biases that can influence clinical decision making.

Given the influence algorithms have on clinical decision making, who then is accountable when a clinical error enters clinical practice? Inevitably, the presence of the AI agent will mean that a clinician will always be influenced by its existence, irrespective of whether the human operator chooses to accept or reject the AI. If the AI's deductions are not to be used by the human operator, but are later discovered to have revealed an outcome that could have prevented the loss of human life, is the human operator accountable for choosing to preserve clinical autonomy and ignore the AI? Inversely, if the AI's conclusions are used by the human operator, but are later discovered to have resulted in the loss of human life due to an unexplained statistical error, is the AI or human accountable and can an AI ever understand the concept of accountability? This is further complicated by the fact that the methods used to increase model interpretability are currently constrained to the discrete case of medical imaging and do little to address concerns around more contemporary unsupervised and reinforcement learning approaches that are increasingly being applied to genomics datasets [110–112], the raw text in electronic health records [60], or in some cases across a mixture of data sources sourced longitudinally across different data contexts [113,114]. In these cases, some ethical issues arise from the fact that even if the computational methods were able to explain themselves, there remains the broader problem that there is no guarantee that we may even be able to understand or validate the conclusion that the AI arrives to.

While these issues are beyond the scope of this paper (refer to [67,84,109] for further discussions), we mention them to highlight that, given the fact that we know that (a) there are several potential sources for algorithmic error and bias, (b) algorithms influence clinical decision making, (c) there is an insensitivity to clinical impact by algorithms, and (d) a lack of guidelines around accountability in the presence of an algorithm; it reinforces our position that a more nuanced socio-technical approach to AI system evaluation prior to clinical integration is necessary if we are to avoid a repeated history where negative unintended consequences arise in yet another HIT integration.

## 6. Further Risks and Limitations from Marginalising Socio-Technical Factors in CRC

In this section, the paper highlights how even if technical and socio-technical concerns are addressed through robust evaluation standards in the interest of patient safety, AI system development and investment should be directed towards problems that have the most measurable impact on patient morbidity and mortality outcomes. Much of the AI optimism in CRC screening has been driven by the fact that it presents a potential solution to the problem that in a routine screening colonoscopy, between 17–28% of colorectal polyps (adenomas) are missed, which is concerning given that for every 1% increase in a clinician's adenoma detection rate (ADR), there is a 3% decrease in the risk of interval cancer [115–117]. Several of the vendor-backed AI augmented diagnostic systems have proposed to address this issue by providing clinicians with a real-time AI polyp detection system, and the evidence provided by recent RCTs is promising. It suggests that the ADR increases by up to 50% with the inclusion of an AI detection system, with the most pronounced effect on trainee gastroenterologists [7–9]. Similarly, in AI-enhanced capsule endoscopy (CE), experimental evidence suggests that AI augmentation consistently outperforms a conventional CE reader in terms of both accuracy and time, showing a 99.88% vs. 74.57% sensitivity in the per-patient detection of abnormalities and, significantly, a 5.9 vs. 96.6-min recording on per-patient reading time between the AI vs. human respectively [10].

While such results are exciting and show some promise in improving patient outcomes, it is also important to recognise that one of the biggest influences on CRC-related mortality is not fundamentally due to the nature of the current technology, but rather due to the prevalence of low rates of participation in CRC screening [118]. Various studies from Australia, which implemented one of the first national approaches to bowel cancer screening, have suggested that at the current participation rate of ~40%, a 15–36% reduction in CRC-related mortality can be expected, and if participation rates in the screening population were to increase to 70%, a 59% reduction in CRC-related mortality would be

observed [119–124]. The problem is that several high-income nations have failed to reach their desired target of 65–80% screening coverage, even in the presence of wide-scale public health campaigns to raise awareness about its importance [81,125]. Unfortunately, it seems that these rates are not likely to increase, given that the rate of FOBT-based screening coverage has plateaued in Australia over the last five years [126]. Participation in follow-up colonoscopy, where many of the AI-enhanced methods are poised to transform outcomes are equally discouraging. Studies from Europe, the United States, Canada, and Australia show that even in the presence of a positive FOBT, only between 50–70% of patients proceed for a diagnostic examination via colonoscopy [126–129]. Participation among the most marginalised groups that have historically experienced social disadvantage, such as those from Indigenous populations, low socio-economic status, cultural and ethnic diversity, or disability, tends to be the lowest in either stage of the screening process [69,130,131].

Consequently, it is possible that the extensive focus of AI adoption and integration into the CRC diagnostic pathway may not have the drastic impact it has promised on patient outcomes. In the following sections, we highlight how the barriers to participation in CRC screening is permeated by human factors, and that if we are sensitive to these factors, we can capitalise on AI methods in a way that can lead to a more significant impact to patient outcomes by developing technology that increases the uptake of screening coverage in high-risk population groups.

### 6.1. Interaction between Patient & Healthcare System

Several international qualitative studies [132–138] have concluded that there are numerous psycho-social and cultural factors that interact and accumulate to impact on a patient's willingness to participate in CRC screening. Barriers include low awareness and a misunderstanding of the medical guidelines around the need for CRC screening and/or believing that screening was only required in the presence of symptoms, which is exacerbated by a lower perception of risk associated with bowel cancer compared with other more high-profile cancers [139]. There has also tended to be limited promotion in community languages among culturally and linguistically diverse populations, which contributes to challenges around the understanding of the purpose of testing and/or how to apply the test kit instructions even if patients choose to proceed [69]. Even where promotion has occurred, screening programme administrators have tended to have limited awareness of how factors, such as culture or gender, influence the way individuals interpret and receive information [132,133]. Additional factors have included the logistics relating to a lack of time to get screened or lack of transportation. For those in urban areas, lack of time could be related to extensive work commitments and a perception of inefficiency by the healthcare system, while for those in rural and remote areas, lack of time may relate to distance and access to healthcare services [134]. Furthermore, fear, anxiety, stigma, shame, uneasiness, or embarrassment in engaging with a procedure that involves stool collection (e.g., the FOBT), or an invasive visualisation of the bowels (e.g., the colonoscopy), both of which may lead to a positive diagnosis of cancer, have been suggested to exacerbate an unwillingness to get screened regardless of sociodemographic context [132].

### 6.2. Interaction between Patient & Clinician

Given the problematic issues around sensitivity and rates of false positives arising from FOBT tests [140,141], patients and clinicians have also been found to convince themselves (in the absence of evidence) that a positive FOBT reading is a false positive attributed to an alternative source of bleeding (e.g., haemorrhoids, menstruation, and straining due to constipation), dietary factors (consumption of beets or orange juice), or medications (e.g., the use of blood thinners). In more extreme cases, some patients more speculatively reported that they believed the toilet was contaminated with someone else's blood, even if the toilet had been cleaned prior [142]. Interestingly, despite the established guidelines around the importance of FOBT-based screening, one qualitative study involving interviews of general practitioner (GP) perceptions of CRC screening in Australia found that

many GPs reinforced negative attitudes towards the FOBT, leading patients to either reject undertaking the FOBT or reject the result of the FOBT in the presence of a positive result. For example, GPs were found to use low risk of bowel cancer arguments to negate the significance of screening, and where a patient returned a positive FOBT, provided explanations that implied that the positive FOBT was more likely the result of a benign source of bleeding [143].

*6.3. Interaction between Clinician & Healthcare System*

System based factors have also been implicated in having a significant role in the management of patients that require CRC screening. Primary care physicians play an important role in the advocacy, facilitation, support, education, and counselling of patients [144]. However, since screening is typically managed outside of the practice setting by a broader national bowel cancer screening infrastructure, there are inconsistencies in the way that results are recorded in patient health records and the availability of that information in the primary care setting. This introduces complexities in the way that GPs flag patients who are overdue for screening, particularly given that FOBT-based screening should occur biennially to achieve its purported benefits. Some studies have suggested that even when the information is made available in the EHR and made accessible in the primary care setting, poor HIT practices have led to workarounds and the under-utilisation of these systems by practitioners [145]. These issues are only compounded in busy primary care settings, where there exists a limited capacity and/or unwillingness to discuss the importance of screening with patients when other more immediate acute and chronic conditions need to be managed in a short consultation session [143].

## 7. The Future of AI and Potential Implications for Clinical Decision Making

In recent years, extensive research and investment has gone into developing (1) novel AI methods that are more emblematic of human reasoning, (2) cloud computing infrastructure to support the storage and retrieval of large volumes of structured and unstructured data through data lakes, and (3) the acceleration of computational processing power in both the domains of super computing and quantum computing. For AI researchers, the belief is that the union of these three fields will result in the holy grail of AI research, artificial general intelligence (AGI). Whether the advancements will correspond to achieving this esoteric goal of human hubris is yet to be seen. However, it does indicate that AI systems are here to stay and both regulators and medical practitioners are likely going to need to grapple with ethical issues surrounding patient autonomy in the presence of contemporary unexplainable AI systems, where some end-of-life patients may argue that they prefer to experiment with an algorithmically inspired personalised therapy that we do not understand, and which may have no evidence.

There is some evidence in the literature to suggest that AI is capable of remarkable feats that can paradoxically be incapable of explanation, problematic for interpretation, yet remain useful to clinical application. The first was an unsupervised learning algorithm known as Deep Patient that was trained on data aggregated from approximately 700,000 patients to broadly predict the health state of an individual by assessing their probability to develop various diseases. The algorithm was evaluated on 76,214 patients comprising 78 diseases across diverse clinical contexts and temporal distributions [60]. Interestingly, the model managed to predict the onset of psychiatric disorders that are notoriously difficult to detect and diagnose by physicians, such as schizophrenia, with remarkable precision, significantly outperforming prior efforts. However, as the lead researcher concedes in an interview, "we can build these models . . . but we don't know how they work" [146]. Similarly, in the field of computational biology, researchers from Google achieved a phenomenal leap forward in the 50-year-old "protein-folding challenge", using an algorithm, AlphaFold, to determine the structure of a protein based solely on its amino acid sequence, achieving an accuracy of 92.4 on the Global Distance Test (GDT) [112]. This has significant ramifications for developing new therapeutics, given that it is the closest

attempt to solving Levintha's paradox, which describes the peculiar situation where there are $10^{300}$ possible configurations of a protein from a typical sequence of amino acids, yet nature folds proteins spontaneously to a consistently exact configuration. The success of such approaches has already found its way into CRC, where deep reinforcement learning has been applied to understand the association between human MicroRNA and colorectal cancer disease progression [147].

It is important to remember that the way AI perceives, interprets, and "senses" reality across millions of data points is epistemologically different to the way humans do, and therefore there may always remain a divide between our understanding of AI and the AI's understanding of the world. Is there a quantitative threshold with which we can "trust" the AI in favour of human judgement in the absence of understanding if the AI is consistently correct across longitudinal observation? This is an open question we currently have no answer to. From the perspective of the EU regulators, a system that cannot explain itself and one that we do not understand has no place in clinical practice. This view is certainly one that seems sensible for the time-being, as it would appear as though the systems that act with profundity are currently the exception, not the rule. However, as more evidence evolves in the near future, it may be the case that persisting with this position may itself be an ethical danger, as preventing individuals from access to personalised algorithmically inspired medical interventions that may be life altering, even in the absence of understanding, will invite new questions around patient autonomy, as some patients may simply prefer to take the risk.

## 8. A Way Forward to Enhancing Clinical Decision Making in CRC: A More Nuanced Approach to AI Systems Development, Implementation, and Evaluation

This paper has presented a socio-technical analysis of contemporary research into the use and impact of AI enhanced HIT in healthcare broadly and in colorectal cancer, specifically to offer a balanced critique on the opportunities, limitations, and risks of AI system development and integration in clinical decision-making. Through this approach, the paper has highlighted socio-technical perspectives on the important contextual nuances that arise from problematic assumptions embedded in the development, implementation, and evaluation of AI systems when applied along the screening, diagnostic, and treatment pathways of CRC.

In Section 3, a series of problematic assumptions underpinning approaches to AI development and implementation in health were identified. These included the general data problems associated with systemic representational biases that manifest elusively through data and the way that unbalanced data distributions tend to marginalise underrepresented groups even in the presence of "big data". An examination of specific issues that permeate supervised learning (intractability of labelling, veracity of labels, and the computational reductionism intrinsic to narrow task definitions), unsupervised learning (the "impossibility" of evaluation and interpretation), and reinforcement learning (selection biases in the framing of environmental data and goals) was then provided. Having examined these issues and in recognition of the fact that minimal external validation of AI in health has existed in both academic research and commercial FDA-approved systems, this paper advocates that, moving forward, it is critical that datasets that are used for ML learning training are independently audited in a transparent way. While the authors do acknowledge that data are perceived as the currency of ML and many vendors have locked commercial agreements in place that are also protected by legislation around patient privacy, it is important to remember that the aircraft industry is supported by an extensive amount of vendor-backed software and hardware components that have managed to cooperate with one another in the interests of safety while maintaining competitiveness. Some researchers, therefore, have advocated that independent auditing analogous to the Aviation Safety Reporting System should exist in healthcare [148].

In Section 4, these issues in the context of AI in CRC were examined and additional nuances were identified in the nature of AI through an evaluation of problematic data

and model assumptions. In data, it was highlighted that there was significant interest in using ML for the purpose of risk stratification and prognostic prediction, particularly given that participation in the existing paradigm of CRC screening was low. However, it was uncovered that several studies reinforced racial biases and used distributions that underrepresented the most marginalised patients, who tend to have the poorest CRC outcomes. It was also observed that class imbalances in modelling the heterogeneity of disease presentation remained a problem, similar to problems that have been identified in the broader AI in health literature. In models, the frame problem of AI was revisited and it was observed that the epistemological and ontological assumptions embedded in ML algorithms were often not resistant to the impact that context-dependent clinical interactions between the patient, clinician, and health system had on clinical decision making. These interactions included the way that knowledge and operational practices evolved over time to create a problem of temporally-influenced model drift; the way that transient effects in the patient circumstances could obfuscate model conclusions in oncologic treatment; the way that spurious cues in data due to situational context could lead to erroneous signal optimisation; the way that operator interactions with AI systems influenced prediction outputs in colonoscopy; the way that a model built for one local populace but commercialised in an international market could not synthesise differences in national guidelines to treatment; and the broader issue where many approaches were not explainable or interpretable.

In examining these issues, the theme that pervaded the multifaceted discussion was that human behavioural studies had identified that clinical decision making had a tendency to be influenced by confirmation bias, algorithmic bias, and automation complacency. Therefore, there was a risk that even small system errors could have major ramifications to patient safety. In recognising these issues, it was identified that the emphasis of the ML literature was on quantitative outcomes, but very few works existed that explored the qualitative socio-technical impact that this would have on patient outcomes. When the issue was discussed, it tended to only ever be conjectured as a potential problem, while studies focused on reporting quantitative outcomes to justify their integration into clinical practice were often not reproduced. This paper therefore suggests that, moving forward, ML studies in health and in CRC have strict standards around: (a) the reproducibility of algorithms prior to publication, (b) reproducibility of quantitative metrics across different socio-organisational settings, (c) a definition for what "interpretability" should look like, given the myriad methodologies that claim to achieve machine interpretability in various ways, and (d) a systems-based approach to the qualitative evaluation that carefully examines: (i) the impact of AI system integration on patient outcomes, (ii) the clinical utility of an AI system, (iii) the nature of AI integration in context to existing HCI and workflow considerations across varied socio-organisational and cultural settings, and (iv) the nature of clinical interaction with and without AI. Further research into a unified quantitative and qualitative methodological framework for AI-enhanced HIT evaluation is urgently required. Indeed, this is something that the broader ML community has identified, motivating an upcoming dedicated conference (ICLR2022 ML Evaluation Standards) to deal with this exact issue [149].

In Section 5, the discussion on AI and clinical decision making was broadened to introduce the fact that there were additional social, ethical, and legal ramifications around the integration of AI systems into clinical practice. It was highlighted that there were problems around the fact that AI itself is insensitive to impact, particularly when probabilistic approaches to clinical decision making that discretise diseases upend decades of medical dogma that has identified health as an interplay that is influenced by complex psycho-social and cultural interactions. Issues pertaining to clinical accountability were also discussed. It was highlighted that the philosophical and ethical debates remain ongoing, particularly around issues of culpability in the presence of machine and/or human error. Given that this research suggests that AI errors are inevitable and will have undue influence on the human observer, moving forward, the authors advocate that careful consideration of the socio-

technical interactions of system use, particularly around the nature of human–computer interaction is examined. The ML model is only one component, whereas how and when the prediction is relayed to the clinician is entirely dependent on the nature of HCI. Therefore, careful examination of how AI is rendered in-use is as critical as the model development, and well-crafted product design solutions inspired by participatory design principles can limit some of the complications that may arise from automation complacency.

In Section 6, the paper reasserted the view that AI efficacy and investment need to be moderated against their impact on patient outcomes. It was identified that the last three decades of technological advancement have done little to perturb the rate of CRC mortality, and that in the presence of increased incidence, it is critical that the one factor that is known to have the most measurable effect on mortality, namely increasing rates of screening participation, should be the context that future AI technology should attempt to address. Complex human factors emerged in the interactions between (a) the patient and healthcare system (misunderstanding, miscommunication, accessibility, cultural sensitivity and broader psycho-social dimensions), (b) the patient and clinician (trust and perception of inefficacy of FOBT), and (c) the clinician and healthcare system (poor interoperability between national and community infrastructure, and poor HIT practices with the EHR). In recognising these issues, the authors emphasise that sensitivity to socio-technical factors in the design and implementation of new technologies is critical. One solution that has not yet been considered in the literature is whether capsule-based technology can be creatively repurposed, reimagined, and repositioned as a tool for the screening of precancerous lesions from home, which may address patient anxieties around (a) the anxiety of a cancer diagnosis, (b) the inconclusiveness of a FOBT result, (c) concerns around the invasiveness of a colonoscopic investigation, and (d) accessibility to healthcare centres both in terms of distance and time.

In Section 7, the paper concluded with a brief look into the future to suggest that the union of big data, high performance computation, and contemporary approaches to unsupervised and reinforcement learning means that regulators may be required to grapple with complex issues around patient autonomy in the era of unexplainable AI. Early evidence suggests that more modern approaches may unlock immense power to the benefit of the patient through personalised medicine. However, this is complicated by the fact that these methods are not interpretable, and even if they were, may never be understood. Where this leaves clinical decision making and clinical autonomy in the presence of patients who may prefer algorithmic clinical decision making needs to be discussed.

## 9. Conclusions

Through a socio-technical analysis of the contemporary literature on AI in CRC, it is evident that a more nuanced approach to AI development and implementation is required. While there is no doubt that AI itself is a transformative technology that has the capacity to positively impact clinical practice to the benefit of the patient, AI optimism needs to be balanced against a thorough understanding of the limitations that also permeate the underlying nature of the technology. This paper highlighted how there are concerns around: (a) biases in end-end data pipelines and technical issues associated with algorithmic model assumptions in the design and development of AI systems; (b) socio-technical issues relating to confirmation bias, automation complacency, interpretability, and the clinician workflow that arises from the interaction with AI systems; (c) ethical and legal implications around accountability and autonomy for both the clinician and the patient; and (d) the potential for misdirected AI investment in the specific context of CRC, where there may be less of an impact on patient mortality and morbidity outcomes, given that the larger issue that proliferates CRC screening is the problem of patient participation that AI currently does little to address. Through the amalgamation of these issues, the authors conclude that the way forward is to develop a more robust mixed methods framework around the auditing and evaluation of AI systems prior to system integration in clinical practice. Such a framework should be guided by principles of data transparency, the reproducibility of

ML models, and more balanced evaluation metrics that weigh quantitative ML metrics against important qualitative clinical considerations, such as (i) the impact of AI system integration on patient outcomes, (ii) the clinical utility of the system, (iii) HCI and clinical workflow considerations across varied socio-cultural and socio-organisational contexts, and (iv) the nature of clinical interaction. In this way, there can be increased confidence that the future of AI in CRC is safe, effective, equitable, and beneficial to clinicians, patients, and the broader health system.

**Author Contributions:** Conceptualization, S.A., P.T., K.-C.Y. and M.-C.W.; methodology, S.A. and P.T.; formal analysis, S.A. and P.T.; investigation, S.A. and P.T.; resources, S.A., P.T., K.-C.Y. and M.-C.W.; data curation, S.A. and P.T.; writing—original draft preparation, S.A. and P.T.; writing—review and editing, S.A., P.T., K.-C.Y. and M.-C.W.; supervision, P.T., K.-C.Y. and M.-C.W.; project administration, S.A., P.T., K.-C.Y. and M.-C.W. All authors have read and agreed to the published version of the manuscript.

**Funding:** This research received no external funding.

**Institutional Review Board Statement:** Not applicable.

**Informed Consent Statement:** Not applicable.

**Data Availability Statement:** Not applicable.

**Acknowledgments:** This research was supported by an Australian Government Research Training Program Scholarship.

**Conflicts of Interest:** The authors declare no conflict of interest.

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
