# Peer review of "AI and Clinical Decision Making: The Limitations and Risks of Computational Reductionism in Bowel Cancer Screening"

_applsci, doi:10.3390/app12073341_

Round 1

Reviewer 1 Report

This paper even though holds interesting insights in AI and ML tools applications and limitations for bowel cancer screening, can improve its structure when it comes to the methodology and results section. The methodology section could include a flowchart of the steps followed while developing the review. Figures could also enhance the results sections. A small conclusion section after section 8 that summarizes the results in only a few sentences would also be recommended. In line 1279, the 150. should be removed.

Author Response

Thank you for your feedback. Please see the attachment for our response.

Reviewer 2 Report

Although my personal opinion differs a lot with respect to what is discussed in the article, I have to accept that it is well written and argued, and therefore, I have no objective reasons for its rejection.

Author Response

(The authors gave the same response as above.)

Reviewer 3 Report

Dear Authors

Thank you for submitting your article to this journal. I believe it is a good review of AI in medical diagnosis and provides a a fairly in depth critique. There are extensive referencing and presentation (English/grammar etc) are fine.

I noted that author contributions section is yet to be completed (line 884). I presume this will be complete in final editing etc.

I believe this paper can be accepted without modification. It is relevant to the journal and it will have a broad audience interested in the development and use of AI in healthcare. 

Although there are already some reviews of AI for healthcare, this  provide further insights and thus merits publication.

Author Response

(The authors gave the same response as above.)

Reviewer 4 Report

This manuscript can't be an article rather a review and socio-technical analysis of AI and healthcare. There is no experiments, no conclusions, no related research section.

Author Response

(The authors gave the same response as above.)

Round 2

Reviewer 4 Report

Thanks to the authors for improving their work. Though, still, I do not see any experiments. I believe that any research article must be accompanied by research experiments.